# Molecular Targets of Oxidative Stress: Focus on Nuclear Factor Erythroid 2–Related Factor 2 Function in Leukemia and Other Cancers

**DOI:** 10.3390/cells14100713

**Published:** 2025-05-14

**Authors:** Syed K. Hasan, Sundarraj Jayakumar, Eliezer Espina Barroso, Anup Jha, Gianfranco Catalano, Santosh K. Sandur, Nelida I. Noguera

**Affiliations:** 1Hasan Lab, Advanced Centre for Treatment, Research and Education in Cancer (ACTREC), Tata Memorial Centre, Navi Mumbai 410210, India; shasan@actrec.gov.in (S.K.H.); anupjha59@gmail.com (A.J.); 2Radiation Biology and Health Sciences Division, Bhabha Atomic Research Centre, Trombay, Mumbai 400085, India; kumarsj@barc.gov.in; 3Department of Life Sciences, Homi Bhabha National Institute, Mumbai 400094, India; 4Santa Lucia Foundation, I.R.C.C.S. Via del Fosso di Fiorano, 00042 Rome, Italy; eliezer90210@gmail.com (E.E.B.); gianfranco.catalano@uniroma2.it (G.C.); 5Department of Biomedicine and Prevention, University of Rome Tor Vergata, 00042 Rome, Italy

**Keywords:** Nrf2, leukemia, metabolism, ROS, NRF2-inhibitors, lipids, glycolysis, PPP, Keap1, AKT-mTOR

## Abstract

Nuclear factor erythroid 2–related factor 2 (Nrf2) is a transcription factor that plays a central role in regulating cellular responses to oxidative stress. It governs the expression of a broad range of genes involved in antioxidant defense, detoxification, metabolism, and other cytoprotective pathways. In normal cells, the transient activation of Nrf2 serves as a protective mechanism to maintain redox homeostasis. However, the persistent or aberrant activation of Nrf2 in cancer cells has been implicated in tumor progression, metabolic reprogramming, and resistance to chemotherapy and radiotherapy. These dual roles underscore the complexity of Nrf2 signaling and its potential as a therapeutic target. A deeper understanding of Nrf2 regulation in both normal and malignant contexts is essential for the development of effective Nrf2-targeted therapies. This review provides a comprehensive overview of Nrf2 regulation and function, highlighting its unique features in cancer biology, particularly its role in metabolic adaptation and drug resistance. Special attention is given to the current knowledge of Nrf2′s involvement in leukemia and emerging strategies for its therapeutic modulation.

## 1. Introduction

The maintenance of redox balance is essential for the normal functioning and health of the cell. Many internal cellular processes, like mitochondrial electron transport, energy metabolism, host cell defense, and external exposure to physical and chemical agents, generate reactive oxygen species (ROS) in the cells. The accumulation of ROS can lead to oxidative stress in the cells and is linked with many diseases in humans [1]. Mammalian cells are equipped with an immense capability to respond to any redox imbalance, and understanding those mechanisms is very important as they have therapeutic implications. The Nrf2 pathway is central for maintaining redox balance and cell cytoprotection. Nrf2’s role in maintaining cellular redox homeostasis, acting as a cancer preventive, is reversed in cancer cells where its function is hijacked to support tumor survival and resistance to therapy [2,3]. Nrf2 is a transcription factor activated in a redox-sensitive manner that transactivates a battery of genes possessing a sequence motif called antioxidant response elements (AREs) in their promoter regions. Nrf2-dependent genes include genes involved in redox homeostasis, cytoprotection and detoxification, drug metabolism, DNA damage response, mitochondrial function, metabolism (of iron, lipids, carbohydrates, and amino acids), etc. [4,5]. The dysregulation of the Nrf2 pathway is observed in many cancers. In a study involving 9125 tumor samples of various cancers, Nrf2 was found to be one of the top ten pathways that have been altered in these tumors, highlighting the importance of Nrf2 pathways in cancer [6]. Nrf2 hyperactivation has been frequently implicated in the development of chemoresistance and radioresistance in cancer cells [7,8]. To harness this pathway for therapeutic intervention, a comprehensive understanding of the regulatory mechanisms controlling Nrf2 activity and how these are disrupted in malignancies is crucial. This review aims to provide a comprehensive overview of Nrf2 regulation and function, highlighting its distinctive features in cancer cells, with a particular emphasis on current knowledge regarding its role and therapeutic modulation in leukemia.

## 2. Nrf2′s Functional Domains and Its Regulation by Keap1

Nrf2 comprises several functional domains that enable its regulation and activity (Figure 1). The Nrf2-ECH homology (Neh) 1 domain is responsible for the DNA-binding activity. It contains a basic leucine zipper (bZIP) motif, enabling Nrf2 to form heterodimers with sMaf (small musculoaponeurotic fibrosarcoma) proteins and to bind ARE promoter sequences of target genes. The Neh 3, 4, and 5 domains are responsible for the transactivation activity of Nrf2, interacting with the transcriptional co-activator CBP/p300, which enhances the transcriptional activity of Nrf2. Kelch-like ECH-associated protein 1 (Keap1), by binding to Nrf2, plays a key role in regulating the functioning of Nrf2 [9]. Nrf2 is subject to complex regulation at the pre-transcriptional, post-transcriptional, and post-translational levels. However, a detailed discussion of these regulatory mechanisms is beyond the scope of this review; there are excellent reviews available for this purpose [10,11]. Keap1-mediated regulation of Nrf2 is the central regulatory mechanism of the Nrf2 pathway. Keap1, bound to Nrf2, functions like a classic two-component system, acting as a sensor and Nrf2 as an effector molecule. The N-terminal region of Keap1 contains the BTB domain, responsible for the dimerization of Keap1 and its interaction with the Cul3-Rbx1-E3 ubiquitin ligase complex [12,13]. Under normal conditions, the Kelch domain found in the C-terminus of Keap1 (composed of six Kelch repeats) binds to the DLG and ETGE motifs present in the Neh2 domain of Nrf2 [14,15]. This binding of Keap1 to Nrf2 leads to the ubiquitination of many lysine residues present in the Neh2 domain of Nrf2 by the Cul3-Rbx1-E3 ligase complex. A specialized protein, p97, binds ubiquitylated Nrf2 from the Keap1–CUL3 E3 complex, leading Nrf2 to proteasomal degradation. The p97 encoding sequence possesses a functional ARE in the promoter, resulting in a negative feedback loop essential for cellular homeostasis under stress conditions [16]. The Keap1 protein also contains an intervening region (IVR) situated between the Kelch and BTB domains, consisting of multiple cysteine residues (Cys151, Cys273, and Cys288) that undergo oxidation by electrophiles [17]. Electrophiles can directly bind to critical cysteine residues, or, under oxidative stress conditions, ROS can act on these key cysteine residues, leading to the disruption of the Keap1-Nrf2 interaction [18]. As a result, Nrf2 is stabilized, accumulates in the nucleus, and activates its target genes. Thus, the IVR and BTB domains function as sensors of oxidative stress, activating the Nrf2 pathway.

The Neh2 domain with DLG and ETGE motifs is recognized by the Kelch domain of Keap1. Cul3-Rbx1-E3 complex binds to the BTB domain of Keap1, involved in the ubiquitination of lysine residues in the Neh2 domain. Some of the critical cysteine residues present in the IVR and BTB domains are indicated. The Neh3, 4, and 5 domains are involved in transactivation function and can bind with the CBP co-activator protein. GSK3β—βTrCP-based regulation happens through the DSGIS motif present in the Neh6 domain. The influence of the PI3K-AKT axis and PTEN on this regulation is also indicated. The Neh1 domain is involved in binding to sMAF and DNA. The Neh7 domain binds with RXRα.

### KEAP-1-Independent Regulation of Nrf2

In addition to the regulation by Keap1, Nrf2 is also regulated through the involvement of other proteins. Apart from ROS and electrophiles, other Keap1 binding partners can disrupt the Keap1-Nrf2 interaction. The autophagy adaptor protein p62 phosphorylation induces the expression of cytoprotective Nrf2 targets under selective autophagic conditions. Its STGE motif is similar to the ETGE and DLG domains present in Nrf2 [19]. The serine residue in this motif can be phosphorylated by the mammalian target for Rapamycin (mTORC1), binding to Keap1, thereby destabilizing the Keap1-Nrf2 complex [20]. Under genotoxic stress conditions, a p53-dependent gene, cyclin-dependent kinase inhibitor1 (p21), can also bind to Nrf2 at the ETGE and DLG domains and stabilize and activate Nrf2 [21,22]. The serine residues present in the DSGIS motif of the Neh6 domain of Nrf2 can be phosphorylated by GSK3β, which increases recognition by β-TrCP and proteasomal degradation by the CUL1-based E3 ubiquitin ligase [23]. GSK3β can, in turn, be inhibited by the PI3K-mediated activation of protein kinase B (AKT) and activated by the PTEN-mediated inhibition of AKT (Figure 1). The Nrf2 pathway can also be regulated by the Neddylation process, in which neural precursor cell-expressed developmentally downregulated protein 8 (NEDD8) is conjugated to CUL3-E3 ubiquitin ligase, leading to the proteasomal degradation of Nrf2 [24].

Nrf2 levels are regulated not only through post-translational mechanisms but also via epigenetic and post-transcriptional processes. At the chromatin level, DNA methylation and histone modifications play critical roles. A prominent example is DNA methylation within CpG islands of the KEAP1 promoter, which is frequently hypermethylated in various cancers, resulting in suppressed KEAP1 expression and the aberrant activation of Nrf2 [25]. Histone modifications also contribute to Nrf2 upregulation; for instance, histone acetyltransferases and methyltransferases such as Mixed-Lineage Leukemia (MLL), which catalyzes H3K4 trimethylation (H3K4me3), have been associated with increased Nrf2 activity and resistance to cancer therapies [26]. The histone acetyltransferase p300/CBP has been shown to directly acetylate Nrf2 at multiple lysine residues within its Neh1 DNA-binding domain. This acetylation alters Nrf2′s promoter-binding specificity, further modulating its transcriptional activity [27]. Post-transcriptionally, Nrf2 expression is extensively controlled by microRNAs (miRNAs) that target either Nrf2 or KEAP1 mRNA. The 3′ untranslated region (UTR) of Nrf2 mRNA contains binding sites for several miRNAs, including miR-144, miR-153, miR-27a, and miR-142-5p, which modulate Nrf2 abundance and activity [28,29]. In leukemia, epigenetic and post-transcriptional mechanisms may contribute to maintaining aberrant NRF2 activation. For instance, promoter hypermethylation reduces KEAP1 expression and increases NRF2 activity. Additionally, KEAP1 mRNA can be directly targeted by microRNAs like miR-200a and miR-144, which further disrupt the KEAP1-NRF2 axis and promote chemoresistance [30]. At the transcriptional level, Nrf2 can undergo modifications that disrupt its interaction with Keap1. In mice, aging brain tissue cells exhibit increased expression of a shorter Nrf2 isoform, which is resistant to KEAP1-mediated inhibition [31]. Similarly, in normal human cells, transcriptional variants that lack exon 2 (containing the DLG region) and are generated by a specific promoter thereby bypassing KEAP1 inhibition. In cancer cells, a similar mechanism can contribute to Nrf2 pathway activation [32,33].

## 3. Regulation of Metabolism by Nrf2

Various genes that are regulated by Nrf2 contain ARE and electrophile response element (EpRE) sequences in their promoter region. For a complete list of genes that are transcriptionally regulated by Nrf2, see Rooney JP et al. [34]. In addition, Nrf2, with its widespread net, is involved in the regulation of key metabolic genes due to its ability to modulate the activity of redox-sensitive metabolic and chromatin-remodeling enzymes. In particular, Nrf2 is involved in promoting chemical reactions that form intermediates for anabolism (anaplerotic metabolism) [35].

Nrf2 directs metabolic reprogramming during stress-inhibiting lipogenesis, supporting fatty acids oxidation (FAO), redirecting glycolysis to the pentose phosphate pathway (PPP), and increasing NADPH regeneration and purine biosynthesis (Figure 2). Such metabolic reprogramming, capable of supporting cell growth and proliferation, is bound to associate with evidence of Nrf2′s key role in the metabolism of many tumors [36]. Several researchers have reported that Nrf2 mRNA is increased in mouse liver by fasting [37,38] and antagonizes hepatic lipid accumulation, suggesting that it contributes to the starvation response during nutrient deprivation [39].

### 3.1. Glycolytic Pathway

Nrf2 induces the expression of several key glycolytic enzymes, including hexokinase 1 and 2 (HK1/2), glucose phosphate isomerase 1 (GPI1), 6-phosphofructo-2-kinase (PFK2), PFK4, fructose-bisphosphate aldolase A (ALDA), enolase 1 (ENO1), and ENO4. The final and rate-limiting step of glycolysis is catalyzed by pyruvate kinase (PK). Among four tissue-specific forms of PK (PKL, PKR, PKM1, and PKM2), the embryonic PKM2 isomer is highly expressed in cancer cells [40] and has been associated with poor prognosis in acute leukemia [41,42]. Junsheng Fu et al. analyzed the metabolomic profiles of esophageal tissues obtained from genetically modified mice that influence the Nrf2 pathway (Nrf2^−/−^, Keap1^−/−^, K5Cre, Pkm2^fl/fl^, Keap1^−/−^, and WT). They assert that, in Nrf2-upregulated cells, PKM2 is upregulated, glycosylated, and oligomerized, resulting in increased glycolytic ATP biosynthesis [43]. On the other hand, in an epithelial environment, ROS accumulation inhibits PKM2 [44,45], diverting glucose into the PPP [43]. Nrf2 also activates the expression of phosphoglucomutase 5 (PGM5), 1,4-alpha-glucan branching enzyme 1 (GBE1), and alpha-glucosidase (GAA), which contribute to glycogen biosynthesis by promoting the conversion of glucose-6-phosphate (G6P) to glucose-1-phosphate (G1P) via PGM. To note that, as is often the case with its pleiotropic regulatory network, Nrf2 upregulation by Keap1 knockout in mouse skeletal muscle cells decreased their glycogen content [46] (Figure 3).

### 3.2. Pentose Phosphate Pathway

Nrf2 enhances the expression of glucose-6-phosphate dehydrogenase (G6PD), the branching point that determines the flux of glucose through the PPP rather than glycolysis [47]. Nrf2 promotes antioxidant defense via the PPP by increasing NADPH availability to regenerate glutathione. An overwhelming mass of reports demonstrates that Nrf2 promotes progression to PPP [48,49,50,51]. Nuclear heme oxygenase 1 (HO-1) interacts with Nrf2, stabilizing its accumulation in the nucleus and inducing preferential transcription of G6PD [52]. In non-neoplastic retinal pigmented epithelial cells, Marisol Cano et al. demonstrated that the expression of G6PD, transaldolase 1 (TALDO1), and transketolase (TKT) was decreased in Nrf2 knockout mice [53].

It has been hypothesized that the increase in PPP activity accounts for Nrf2-dependent proliferation, as PPP genes are strongly activated by Nrf2 in proliferating cells where the phosphatidylinositol-3-kinase (PI3K)/protein kinase B (AKT) pathway is active [54]. Moreover, it has been demonstrated that in zebrafish hepatocyte injury recovery, Nrf2 activation is required to induce the PPP, and inhibition of the PPP disrupts nucleotide biosynthesis, thereby preventing liver regeneration [55] (Figure 4).

### 3.3. Purine Biosynthesis Pathways

Nrf2, a regulator of anaplerotic metabolism, activates anabolism genes, promoting the proliferation and survival of normal and cancer cells. Metabolomic profiling revealed that Nrf2 promotes purine nucleotide synthesis and glutamine metabolism in the presence of active PI3K-Akt signaling. Nrf2 activates transcription of genes related to de novo nucleotide synthesis, phosphoribosyl pyrophosphate amidotransferase (PPAT), methylenetetrahydrofolate dehydrogenase 2 (MTHFD2), genes encoding enzymes for NADPH synthesis, malic enzyme 1 (ME1), and isocitrate dehydrogenase 1 (IDH1) [54]. PPAT transfers an amine group from glutamine to phosphoribosyl pyrophosphate (PRPP) in the first step of purine synthesis while MTHFD2 is essential for the synthesis of the 10-formyl-tetrahydrofolate molecules to be added to the purine ring [41].

### 3.4. Lipid Metabolism

Keap1 disruption and Nrf2 activation in mouse skeletal muscles enhance fatty acid β-oxidation, mitochondrial activity, and energy production during exercise [56]. Nrf2 is also key in adipogenesis, regulating preadipocyte differentiation through PPARγ and C/EBPα/β [57,58,59,60,61,62]. Additionally, Nrf2 counteracts hepatic lipid accumulation, aiding starvation response [39] and mediating cytoprotective gene induction in fetal liver under caloric restriction [63].

Nrf2 may also influence cholesterol synthesis by preventing 3-hydroxy-3-methylglutaryl-CoA (HMG-CoA) reductase inactivation, which catalyzes the rate-limiting step in cholesterol biosynthesis [47].

With aging, reduced antioxidants and increased ROS in Leydig cells impair testosterone production; however, sulforaphane-induced Nrf2 activation restores this balance, helping in steroid production [64]. The action of Nrf2 is also involved in acute myeloid leukemia (AML) therapy resistance related to lipid metabolism. Picou et al. showed that n-3 polyunsaturated fatty acids (PUFAs), present in fish oil at high concentrations, have anti-tumoral properties, leading to cell death through the disruption of the mitochondrial metabolism of AML cells, associated with oxidative stress and Nrf2 response [65]. A ChIP-Seq analysis highlights Nrf2′s regulation of key lipid metabolism genes, including the elongation of very long-chain fatty acids protein 7 (*ELOVL7*), acyl-CoA synthetase, short-chain family member 1 (*ACSS1*), acyl-CoA thioesterase 7 (*ACOT7*), fatty acid desaturase 1 (*FADS1*), acyl-Coenzyme A dehydrogenase family member 10 (*ACAD10*), acyl-Coenzyme A dehydrogenase family member 12 (*ACAD12*), lipase member H (*LIPH*), acetyl-CoA oxidase 1 and 2 (*ACOX1* and *ACOX2*), phospholipase A2 (*PLA2G7*), and patatin-like phospholipase domain containing 2 (*PNPLA2*) [43,66]. Nrf2 also activates CD36, modulating lipid uptake [67,68]. It influences CPT1 and CPT2, enzymes essential for mitochondrial fatty acid transport [69,70,71], downregulate ATP citrate lyase, very long-chain fatty acid elongase 6 (ELOVL6), and stearoyl-CoA desaturase, limiting fatty acid synthesis [72,73]. Murine studies show that Nrf2 overexpression alleviates hepatic lipotoxicity by suppressing lipogenic genes like acetyl-CoA carboxylase 1 (ACC1) and fatty acid synthase (FASN) [74]. Taken together, these findings suggest that Nrf2 enhances fatty acid β-oxidation while inhibiting lipid biosynthesis, with dysregulation linked to metabolic diseases like cancer. In HepG2 cells, Nrf2 fosters lipid accumulation by upregulating stearoyl-CoA desaturase-1 (SCD1), ACC1, FASN, sterol regulatory element-binding protein 1 (SREBP-1), and PPAR-α, an effect amplified by Nrf1 [75] (Figure 5).

### 3.5. Amino Acid Metabolism

Amino acids (AAs) play a crucial role in the redox balance of cells. Glutamine, glutamate, and aspartate stimulate the activity of Nrf2 [76], while cysteine serves as a precursor in antioxidant synthesis and directly modulates NRF2 activity [77]. In turn, Nrf2 regulates AA availability via activating transcription factor 4 (ATF4) expression, increasing glutamine transporter solute carrier family one member 5 (SLC1A5) expression for glutamine uptake and enhancing LAT1 and ASCT2 transporter activity [54].

Glutamine, the most abundant AA in plasma, is crucial for nucleotide and nonessential AA biosynthesis. Nrf2 promotes glutaminase (GLS2) and glutamic pyruvate transaminase 2 (GPT2), driving glutaminolysis to generate glutamate, aspartate, alanine, and α-ketoglutarate, key intermediates for metabolism [43]. Glutamine-derived α-ketoglutarate fuels energy production and TCA cycle replenishment, supporting cancer cell survival [54]. Targeting glutamine metabolism has antileukemic potential [78,79,80,81,82].

As mentioned previously, Nrf2 also regulates PKM2, influencing glucose flow into serine metabolism, which supports cysteine, glycine, and methionine synthesis via one-carbon metabolism, crucial for protein and nucleic acid biosynthesis [83,84,85] (Figure 6).

## 4. Nrf2-Regulated Mitochondrial Functions

This comprehensive metabolic control by Nrf2 is intimately connected with mitochondrial function, given that mitochondria serve as both a metabolic hub and a major source of reactive oxygen species (ROS). Nrf2 not only responds to mitochondrial ROS (mROS) but also influences mitochondrial biogenesis, redox homeostasis, and energy metabolism.

Nrf2 reciprocity with mitochondria and its metabolism is well documented, though not all underlying mechanisms have been fully elucidated. ROS production induced by mitochondrial metabolism (mROS) and Nrf2 pathway upregulation is interdependent and influenced by mitochondrial bioenergetics. It is widely accepted that excessive ROS production from mitochondrial metabolism activates Nrf2, although the precise regulatory pathway remains unclear.

According to Alam et al., Nrf2 mitochondrial activation may result from enhanced sulfur oxidation due to the upregulation of sulfide quinone oxidoreductase (SQOR), which supports mitochondrial function and cellular redox balance [86]. However, limited reports demonstrate a direct mechanism by which mROS activates Nrf2 [87,88]. Since mitochondrial superoxide has a short half-life, its effects may be mediated by conversion to hydrogen peroxide, which can diffuse into the cytosol and subsequently activate Nrf2 [89].

Cano et al. demonstrated that Nrf2-deficient retinal pigmented epithelial cells exhibit increased H_2_O_2_ production, leading to impaired mitochondrial function. Although Nrf2 does not directly affect mitochondrial antioxidant abundance, its deficiency elevates levels of oxidized Peroxiredoxin 3 (PRX3) due to decreased NADPH synthesis from IDH2 and PPP genes [53].

Kovac et al. reported that mitochondrial ROS levels were elevated in cells from Nrf2-KO mice compared to their wild-type counterparts, as mitochondrial ROS production is regulated by the Keap1–Nrf2 pathway through its control over mitochondrial bioenergetics. Interestingly, Keap1-KO cells also displayed increased ROS production, albeit to a lesser extent, due to differential expressions of NOX genes (NOX2 is upregulated in Nrf2-KO, whereas NOX4 is upregulated in Keap1-KO cells with constitutive Nrf2 activation) [90]. Similarly, in Nrf2-deficient mice, brain mROS production is upregulated due to impaired mitochondrial respiration. Constitutive Nrf2 activation in Keap1-KO mouse brains also led to elevated mROS levels, potentially linked to mitochondrial hyperpolarization.

Supplementing excess pyruvate to feed the TCA cycle and enhance NADH production significantly reduced ROS levels in Nrf2-KO neuronal–astrocytic co-cultures, confirming a fuel deficiency for complex I [89,90]. In various cellular environments (lung and myocytes), preventing Nrf2 binding to Keap1 upregulates genes encoding proteins of the respiratory complexes (ubiquinone, complexes I-IV, ATP synthase subunits, and mitochondrial pyruvate carrier) and enzymes of the TCA cycle, glucose, and fatty acid metabolism [70,91,92]. Conversely, Nrf2 inhibition downregulates proteins linked to mitochondrial respiration, including a subunit of complex I encoded by the *NDUFS6* gene and key metabolic enzymes such as isocitrate dehydrogenase (IDH), pyruvate dehydrogenase phosphatase regulator (PDP), and long-chain-specific acyl-CoA dehydrogenase (LCAD) [47,92].

Additionally, Nrf2 may indirectly regulate mitochondrial pathways by preventing oxidative-stress-induced modifications to glucose metabolism, the TCA cycle, and the electron transport chain (ETC) [47,93]. Transgenic mouse models demonstrate Nrf2′s regulatory role in mitochondrial respiration. Holmström et al. found that in Nrf2-KO cells, respiration was impaired, leading to lower mitochondrial membrane potential and reduced ATP production [94]. This impairment stemmed from reduced NADH and FADH_2_ substrate availability for ETC complexes I and II. The inefficient oxidative phosphorylation (OxPhos) in Nrf2-KO neurons resulted in lower ATP levels and increased glycolytic dependence. Conversely, genetic Nrf2 activation enhanced respiration rates and substrate availability, promoting higher ATP production via OXPHOS and elevating mitochondrial membrane potential [94].

Nrf2 enhances mitochondrial substrate availability by stimulating TCA cycle activity. Consistent with this, Singh et al. demonstrated that Nrf2 regulates carbon flux through the TCA cycle [95]. Cvetko et al. suggested that the depletion of mitochondrial GSH and the inhibition of its thioredoxin system might be key mechanisms leading to Nrf2 activation and supporting mROS production [96]. Complex I activity can independently induce Nrf2 through fumarate accumulation, regardless of mROS levels [97].

A ternary complex comprising PGAM5 (PGAM family member 5), Nrf2, and Keap1 on the outer mitochondrial membrane (OMM) may act as a sensor of mitochondrial function and ROS production. Disrupting this complex enhances Nrf2 activity, highlighting its crucial role in maintaining mitochondrial health [98]. Nrf2 also protects mitochondrial function by preventing the mitochondrial permeability transition pore (mPTP) opening in response to stress. In lipopolysaccharide-treated microglia, Nrf2 activation mitigated mPTP induction and safeguarded against apoptosis [99].

Hoang et al. showed that in leukemia stem cells (LSCs), arsenic trioxide (ATO)-induced ROS production was enhanced when combined with the Bcl2 inhibitor Venetoclax (VEN). VEN disrupted ATO-induced Nrf2 translocation, amplifying ATO-driven ROS levels and increasing apoptosis in LSCs. The ATO + VEN combination reduced mitochondrial membrane potential, mitochondrial size, FAO, and oxidative phosphorylation, collectively enhancing apoptosis in LSCs derived from both VEN-sensitive and VEN-resistant AML primary cells [100].

## 5. Nrf2 Dysregulation in Cancer

A study analyzing the alteration patterns of over 9000 tumor samples revealed that the Nrf2 pathway is frequently dysregulated across multiple tumor types. Among various tumor types, the study identified alterations in the Nrf2 pathway in 25% of lung squamous cell carcinomas, 23% of stomach and esophageal cancers, 19% of uterine corpus endometrial carcinomas, and 15% of lung adenocarcinomas [6]. Most of these alterations involve gain-of-function mutations and amplifications in Nrf2, as well as loss-of-function mutations in its negative regulators, Keap1 and CUL3 [101]. The dysregulation of the Nrf2 pathway in cancer can be attributed to multiple factors, including mutations in Nrf2 or its associated regulatory genes, epigenetic modifications, and changes in microRNA expression. Additionally, the tumor microenvironment, characterized by oxidative stress and inflammation, can contribute to Nrf2 activation in cancer cells. The role of Nrf2 in cancer is complex and context-dependent, and its precise effects may vary depending on the type of cancer and stage of the disease [102,103,104]. Nevertheless, Nrf2 dysregulation has emerged as a promising therapeutic target in cancer, and several drugs targeting Nrf2 or its downstream pathways are currently under development [104,105].

### 5.1. Somatic Mutations in Nrf2-Keap1 System in Cancers

Somatic mutations in the Nrf2-Keap1 system have previously been reported in many cancers, including lung, esophageal, hepatocellular, head, and neck carcinoma [104,106,107]. Missense mutations in Keap1 lead to the accumulation and activation of Nrf2 [13]. These mutations target key cysteine residues in the Keap1 protein, resulting in its inactivation and subsequent stabilization of Nrf2.

According to Hast et al., KEAP1 mutations, including N469fs, P318fs, R554Q, W544C, and G333C, disrupt Keap1-Nrf2 binding in lung squamous cell carcinoma. In contrast, mutations such as R470C, S243C, G423V, G186R, V155F, R320Q, and D422N, despite functioning as “super-binder” variants, fail to suppress Nrf2 activity [108]. Although they are less frequent than Keap1 mutations, mutations in Nrf2 itself have also been found in certain malignancies. Through the ETGE and DLG motifs, two conserved motifs found in Nrf2′s N-terminal tail, Keap1 interacts with Nrf2 and these motifs have been identified to contain somatic mutations of Nrf2 in cancer. Kerins et al. performed an exhaustive analysis of Nrf2 somatic mutations in the TCGA database for 10,364 tumor cases, spanning 33 tumor types, and found Nrf2 mutations in 21 of 33 tumor types. Non-synonymous mutations at amino acid positions W24, Q26, D29, L30, G31, R34, D77, E79, T80, G81, and E82 were found to be overrepresented across all mutation-containing tumor types. Certain mutations such as W24, Q26, R34, and D77 were present outside of the ETGE and DLG motifs. However, R34 was observed to be the most frequently mutated residue of Nrf2 (14.2% of all Nrf2 mutations) [109]. These mutations can lead to the constitutive activation of Nrf2, independent of Keap1 regulation. Depending on the type, location, and cellular context of the mutation, Nrf2-Keap1 mutations in cancer may have different particular effects. But in general, these genetic alterations are believed to support tumor growth and survival via improving cellular detoxification and antioxidant pathways, as well as by inhibiting cell cycle arrest and apoptosis [108,109].

The identification of Nrf2-Keap1 mutations in cancer has led to interest in developing targeted therapeutic strategies, such as small molecule inhibitors of Nrf2 or Keap1 [105,108]. However, the development of such therapies has been challenging, as the Nrf2 pathway is also critical for normal cellular homeostasis and functions. Therefore, careful consideration of potential side effects and toxicities will be necessary in the development of Nrf2-Keap1-targeted therapies for cancer.

### 5.2. Metabolic Dysregulation in Cancer

As discussed in the sections above, Nrf2 is a key regulator of cellular stress-induced metabolism. Through its direct or indirect actions, Nrf2 facilitates metabolic pathways that support cell proliferation and may contribute to cancer development and progression by modulating cellular metabolism. For instance, the inhibition of G6PD, a pathway upregulated by Nrf2 via HO-1, has shown anti-tumorigenic effects in leukemia, glioblastoma, and lung cancer cells [110].

In cancer cell lines with constitutive Nrf2 accumulation (A549, H2126, LK2, and EBC1 cells), Yoichiro Mitsuishi et al. showed that Nrf2 activates genes whose products are involved in TKT, phosphogluconate dehydrogenase (PGD), G6PD, TALDO1, and PPP [54]. Furthermore, by reducing the expression of miR-1 and miR-206, Nrf2 can increase the expression of the PPP genes, improving PPP-dependent NADPH synthesis and stimulating the proliferation of tumor cells [95]. When breast tumors recur, Nrf2 promotes de novo nucleotide synthesis and regulates redox homeostasis [36]. An in vivo CRISPR screening demonstrated that Nrf2 promotes 6-phosphogluconate dehydrogenase, the third enzyme of the oxidative PPP, and PPAT, both required for tumor growth [111]. It has been proven that the pluripotency of leukemic stem cells and self-renewal were promoted by the PIK3γ-AKT axis through the induction of the PGD and PPP pathway mediated by Nrf2 [112], further highlighting the importance of Nrf2 in directing metabolism in cancer cells [85].

Obesity in leukemia patients has been associated with poor prognosis, suggesting that a lipid-rich environment could be protective for leukemic stem cells (LCSs) [113]. Adipocytes are prevalent in the bone marrow (BM) stroma and increase in number with age (15% of the BM in 20-year-old patients and 60% in 65-year-old patients). Adipocytes supply fatty acid (FA) ligands to leukemia cells that induce PPARγ-regulated FA oxidation genes, thereby promoting cell survival. It is plausible that quiescent and undifferentiated LSCs might derive most of their energy from fatty acid oxidation (FAO) activation, which could make them vulnerable [114]. CD36 facilitates long-chain fatty acid uptake into cells; its overexpression is associated with poor prognosis and shorter overall survival in AML patients [115] and cancer metastasis [116,117]. According to Gregory Ma et al., CB-839 inhibits glutamine catabolism and reduces the amount of glutamate available in the cell for the synthesis of GSH and α-ketoglutarate (αKG), which serves as a carbon source for the TCA cycle and other metabolic intermediates. This led to the impairment of redox metabolism in AML cells, causing mitochondrial ROS accumulation by limiting the synthesis of antioxidant glutathione, resulting in apoptosis. CB-839 makes AML cells more sensitive to antileukemic drugs that induce mitochondrial oxidative stress, like ATO and homoharringtonine (HHT) [118].

Dan Huanga et al., patenting an endogenous fluorescent probe for the screening of H_2_O_2_ in transgenic AML murine models, were able to distinguish two populations of leukemic cells, with a low and high production of H_2_O_2_. The population with low-H_2_O_2_ had improved leukemogenic capacities; much lower levels of ATP, Δψ_m_, succinate, and alpha-ketoglutarate; but higher levels of NADPH, pyruvate, and lactate. Transfecting MLL-AF9 (low-H_2_O_2_ phenotypes) into recipient mice, they found that the development of leukemia in these was faster compared to leukemia models transduced with granulocyte-macrophage progenitors (GMPs) or hematopoietic stem cells (HSCs), thus indicating that redox metabolism (but not the pluripotency state) may be important in promoting leukemogenesis. Furthermore, the low-H_2_O_2_ population had greater resistance to chemotherapy, related to the overexpression of malic enzymes ME1/3, which are in part transcriptionally regulated by Nrf2 [119].

### 5.3. Nrf2 Regulatory Networks in Leukemia

In leukemia, the dysregulation of Nrf2 and its downstream pathways has been reported, and several studies have investigated the Nrf2 regulatory networks in this disease [120]. Analyzing data from the GEO: GSE30029 dataset, we observed a significantly higher level of *Nrf2 mRNA* in CD34- bulk AML cells and even in CD34+ AML cells, representing the leukemic stem cells population, than in normal CD34+ cells. (Figure 7A). Since Nrf2 protein levels and functions are tightly regulated in a post-transcriptional fashion, mRNA-level data need to be carefully pondered. We previously demonstrated that promyelocytic leukemia/retinoic acid receptor alpha (PML/RARa) hybrid protein inhibits Nrf2 function, impedes its transfer to the nucleus, and enhances its degradation in the cytoplasm, but mRNA levels are similar to those found in other AML subtypes. Such a loss in Nrf2 function alters cell metabolism, demarcating APL tissue from both normal promyelocytes and other AML blast cells [121]. Considering TCGA database’s data, we could observe no difference in overall survival among AML patients based on Nrf2 mRNA expression (Figure 7B), but searching for the presence of Nrf2 mutations, we could see that AMLs with an altered Nrf2 show a lower overall survival compared to the wild-type group (*p* = 0.02) (Figure 7C). Such data suggest that an Nrf2 mutation may be associated with poor survival.

In AML, Nrf2 activation has been shown to promote cell survival and resistance to chemotherapy. Chemotherapy has been found to induce Nrf2 activation by relieving its Keap1-mediated negative regulation in AML. Additionally, Nrf2 activation in AML cells can promote the expression of BCL-2 and BCL-XL, while downregulating BAX and caspase 3/7, further contributing to cell survival [100]. Hu et al. found that Nrf2 was overexpressed in relapsed and refractory AML patients who harbor genetic mutations: Nrf2 interacted with the promoter region binding site of replication factor C4 (RFC4), an important component of the DNA mismatch repair (MMR) pathway, thus attenuating its activity. They observed that enhanced p65/c-Jun/JNK signaling contributes to drug resistance in those AML patients [122].

Some studies have also elucidated the contribution of Nrf2 in promoting cellular differentiation in AML cells. Certain compounds, such as carotenoids, polyphenols, etc., stimulate Nrf2 release from Keap1-mediated negative regulation, eventually causing the transactivation of the ARE, which are found in the promoter regions of genes that encode phase II detoxifying and antioxidant enzymes as well as proteins having a reducing ability like peroxiredoxin and thioredoxin [100,122]. Nrf2 enhances the differentiation-promoting ability of carnosic acid (CA, a plant-derived polyphenolic antioxidant) when co-administered with 1α,25-dihydroxyvitamin D_3_ (1,25D) in AML cells via the upregulation and activation of activator protein-1 (AP-1) and the vitamin D receptor [123]. Nrf2 is involved in the upregulation of several detoxification and cytoprotective genes in response to chemotherapeutic drugs and also in limiting apoptosis, thereby mediating drug resistance in AML patients [124]. In acute promyelocytic leukemia (APL), Nrf2 has been found to be involved in the control of cellular sensitivity to arsenic trioxide (ATO), a therapeutic agent used in refractory and recurrent APL patients. Nishimoto et al. found that the pretreatment of NB4 cells with CA resulted in increased Nrf2 activity, thereby elevating the levels of multidrug resistance protein 1 (MRP1), which enhanced arsenic efflux before ATO-induced apoptosis [125].

A study from Bonovalis et al. elucidates the involvement of Nrf2 targets HO-1 and NQO-1 in resistance to imatinib in chronic myeloid leukemia. Nrf2 was found to be overexpressed in an imatinib-resistant K562 CML cell line. It was also found to be significantly positively correlated with thioredoxin reductase (TrxR) expression. Nrf2 knockdown in such cells restored imatinib sensitivity [126]. An increased HO-1 level induces AP-1 expression and enhances NF-kB signaling, both of which lead to increased leukemic cell survival [127]. Wang et al. demonstrated that Nrf2 and its target protein, glutathione-S-transferase alpha (GST-α), are overexpressed in imatinib-resistant K562 cells [128]. Increased GST-α was found to be involved in maintaining cell survival by negatively regulating intracellular levels of 4-Hydroxynonenal (4-HNE). Additionally, increased TrxR and GST-α levels have also been implicated in driving imatinib resistance by enhancing the activity of the BCR-ABL1 fusion oncoprotein.

Nrf2 mutations have also been found to be involved in mediating drug resistance in acute lymphoblastic leukemia (ALL). Akin-Bali et al. have demonstrated the presence of 6–8 novel mutations in the Nrf2-Keap1 signaling pathway by performing an in silico study on 30 ALL patients. Using a PolyPhen-2 program analysis, they were able to identify several pathological mutations in the NF-kB1, KEAP1, and p62 signaling pathways [129]. PAQR3 has been found to suppress cell proliferation by negatively regulating Nrf2 levels. In acute lymphoblastic leukemia, the overexpression of Nrf2 has been found to inhibit the anti-proliferative activity of PAQR3 [130]. The Nrf2 inhibitor brusatol has anti-tumor activity in in vitro models of B-ALL and T-ALL [131]. Wang et al. correlated the role of Nrf2 overexpression with decreased sensitivity to vincristine chemotherapy in B-ALL [132]. Chemotherapy induced the increased activation of the Nrf2 and PI3K/Akt signaling pathway in B-ALL. Elevated Nrf2 levels negatively regulated BAD levels, thereby decreasing apoptosis and creating a resistance loop. Studies on REH and MOLT-4 ALL cell lines by Gajda et al. uncovered that the inhibition of mitogen-activated protein kinase (MAPK)/extracellular signal kinase (ERK) and PI3K/AKT signaling pathways led to a decrease in the levels of Nrf2/NF-kB and the prevention of drug resistance in these cells [133]. In chronic lymphocytic leukemia (CLL), Nrf2 activation can enhance the expression of DNA repair genes such as XRCC1 and POLB, which may contribute to disease progression and resistance to chemotherapy. Several regulatory networks have been implicated in Nrf2 dysregulation in leukemia [134,135]. For example, aberrant activation of the Wnt/β-catenin signaling pathway has been shown to upregulate Nrf2 expression in AML cells. Similarly, dysregulation of the PI3K/AKT pathway can also lead to Nrf2 activation and promote leukemia cell survival [135]. BAFF released from monocyte-derived nurse-like cells (NLCs) present in the tumor microenvironment enhances the accumulation of p62/SQSTM1, which triggers Nrf2 activation in CLL cells. CLL patients report high BRD9 levels, which have been found to transcriptionally activate Nrf2 expression. Enhanced Nrf2 expression and activation have been found to cause resistance to ROS-modulating therapeutics such as venetoclax and cause cell survival through increased mTORC1 signaling [136]. In addition, alterations in microRNA expression have been reported in leukemia, and some of these microRNAs can target Nrf2 and its downstream pathways. For example, miR-153 has been shown to inhibit Nrf2 expression in some AML cells, leading to increased sensitivity to chemotherapy [137]. Overall, the Nrf2 regulatory network in leukemia is complex and context-dependent, and its precise effects may vary depending on the type and stage of the disease, as well as other genetic and environmental factors. However, knowledge of the Nrf2 regulatory networks in leukemia may help identify possible treatment targets for the disease (see Table 1 and Figure 8).

## 6. Nrf2 and Cancer Treatment Resistance

The primary challenge that arises during cancer treatment is the emergence of resistance against chemotherapy and radiotherapy. Nrf2 mediates the induction of AREs containing genes that actuate cellular defense against oxidative stress and make active molecules and metabolites water soluble and easily excretable by the cells; thus, it is a transcription factor vital to chemoprevention, but also a vital factor for cancer cells challenged by therapy. It has been observed that Nrf2 is dysregulated and hyperactivated in multiple cancers. Some of the known mechanisms driving this hyperactivation are due to mutations in *NRF2*, *KEAP1*, or *CUL3*; the epigenetic silencing of *KEAP1*, *RBX-1,* or *CUL3;* and the amplification of *NRF2*. Interestingly, most of the gain-of-function mutations observed in Nrf2 are located at the DLG and ETGE motifs. In cancer cells, Nrf2 is known to play an important role in sustained proliferation, resisting apoptosis, metabolic reprogramming, and modulating redox homeostasis [134]. Since Nrf2 is involved in these crucial functions, it also imparts a development of resistance to chemotherapeutic drugs and radiotherapy. A CRISPR-Cas9 deletion screen identified Nrf2 activation mediated by KEAP1 knockout as a primary factor that confers resistance to many chemotherapeutic drugs in lung cancer cells. This resistance was attributed to an Nrf2-mediated increase in glutathione synthesis and decreased drug-induced ROS and altered the metabolism of the cells so that the cells could grow in the absence of MAPK signaling [139]. Nrf2 is a master regulator of redox homeostasis by governing the expression of an array of antioxidant genes such as glutathione pathway genes (GCLC, GCLM, glutathione reductase, and glutathione peroxidase), thioredoxin pathway genes (NADPH synthesis, thioredoxin1 and 2, thioredoxin reductase 1 and 2, and peroxyredoxins), heme oxygenase-1, etc. Many chemotherapy drugs and radiation act by generating ROS in the cells, and these hyperactivated antioxidant enzymes can scavenge/detoxify ROS, leading to the development of resistance by cancer cells. Nrf2 regulates a broad spectrum of genes essential for drug efflux and detoxification, contributing significantly to cellular defense mechanisms. Among the best-characterized targets involved in drug efflux are ABCC1, ABCC2, ABCC3, and ABCG2, which encode ATP-binding cassette (ABC) transporter proteins [7]. These transporters actively extrude chemotherapeutic agents from cells, thereby reducing intracellular drug accumulation and diminishing therapeutic efficacy. In parallel, Nrf2 induces the expression of numerous detoxification enzymes, including NQO1 (NAD(P)H:quinone oxidoreductase 1), glutathione S-transferases (GSTs), UDP-glucuronosyl transferases (UGTs), and HMOX1 (heme oxygenase-1), which participate in oxidoreduction, hydroxylation, and conjugation reactions to neutralize and eliminate toxic compounds [140]. The constitutive activation of Nrf2 in cancer cells enhances both drug efflux and detoxification pathways, ultimately contributing to resistance against a wide range of therapeutic agents. As stated above, constitutively active Nrf2 also accelerates the efflux of drugs from the cells and the inactivation or detoxification of drugs, thereby making cancer cells refractory to treatment. Hyperactivated Nrf2 can also hamper the induction of apoptosis through activating many antiapoptotic genes like *BCL2*, *BCL2L1*, and *HIPK2* [141]. Many chemotherapeutic drugs and radiation kill cancer cells by causing DNA damage. Nrf2 is known to activate DNA repair genes like *TP53BP1* [142,143], *RAD51*, and 8-oxoguanine DNA glycosylase (*OGG1*) in cancer cells [144]. Many other genes that are involved in DNA repair, such as *RAD52*, *XRCC2*, *XRCC3*, *DMC1*, *RBBP8*, and *SHFM1*, also contain AREs in their promoter regions [144]. The Nrf2-mediated hyperactivation of DNA repair proteins is bound to make the cells refractory to chemotherapy and radiotherapy, leading to the development of resistance. Nrf2 also contributes to resistance to chemotherapeutic drugs (Cytarabine and daunorubicin) in AML [145]. Nrf2 activation has been linked to the emergence of clones harboring the arginine 882 mutation in the DNA methyltransferase 3A (*DNMT3A*) gene, leading to daunorubicin resistance in many AML patients [30]. Radiotherapy and chemotherapy are known to cause the activation of Nrf2 even in tumor cells where the Nrf2 pathway is not mutated or inherently dysregulated [146]. Due to this induced Nrf2 activation, cancer cells can acquire resistance.

Beyond its established roles in detoxification and drug efflux, Nrf2 also contributes to treatment resistance by regulating autophagy. While autophagy can act as a tumor-suppressive mechanism under certain conditions, growing evidence suggests that it frequently supports cancer cell survival and resistance to therapy [147,148]. Nrf2 directly modulates autophagy by upregulating the expression of key autophagy-related genes, including *SQSTM1/p62*, *ULK1*, *ATG5*, and *GABARAPL1* [149]. This regulation establishes a positive feedback loop, as p62 facilitates the selective degradation of Keap1 via autophagosomes, further enhancing Nrf2 activity [150]. Consequently, targeting the Nrf2–autophagy axis represents a promising strategy for overcoming therapeutic resistance. However, the cytoprotective effects of autophagy vary depending on cell type, stress conditions, and tumor context, underscoring the need for further investigation into the specific role of this pathway across different malignancies.

Enhanced stemness is another major mechanism underlying drug resistance in cancer. Aldehyde dehydrogenase 1 (ALDH1A1), a recognized cancer stem cell (CSC) marker, is linked to self-renewal capacity and resistance to chemotherapy. Nrf2 regulates ALDH1A1 expression, sustaining stemness in several cancers, including breast, prostate, and pancreatic tumors. Additionally, Nrf2 indirectly modulates the expression of key stemness-related genes such as CD44 and NOTCH by mitigating oxidative stress, further reinforcing drug resistance. A recent study revealed that ZMYND8, a histone reader protein implicated in therapy resistance in breast CSCs, physically interacts with Nrf2, enhancing the transcription of Nrf2 target genes and protecting CSCs from ferroptosis [151]. Notably, ZMYND8 has also been identified as a direct transcriptional target of Nrf2, suggesting the existence of an additional positive feedback loop that reinforces Nrf2 signaling in CSCs [151]. The epithelial–mesenchymal transition (EMT) also contributes to treatment resistance and poor clinical outcomes in cancer [152]. Nrf2 has been shown to promote the EMT and stabilize a hybrid epithelial/mesenchymal (E/M) state, thereby increasing cellular plasticity and resistance to therapy [153]. This effect is mediated both through the direct upregulation of EMT transcription factors such as *SNAIL* [154] and through the modulation of key EMT-inducing signaling pathways, including TGF-β, Notch, Wnt, and HIF-1α [155,156,157]. Given these multifaceted roles, the inhibition of Nrf2 may reverse the EMT and stemness phenotypes, thereby enhancing tumor sensitivity to conventional therapies. Hence, Nrf2-mediated drug resistance has become a significant clinical challenge in cancer treatment. The identification of the underlying mechanisms and the development of strategies to overcome Nrf2-mediated drug resistance are essential for the effective management of cancer. The development of suitable Nrf2 inhibitors will be key to overcoming this treatment resistance.

### Role of Nrf2 in Ferroptosis-Mediated Drug Resistance

Ferroptosis is a form of cell death caused by lipid peroxidation due to increased intracellular labile iron pool (LIP) and ferroptosis is distinct from necrosis, apoptosis, and autophagy [158,159,160,161]. Nrf2 inhibits ferroptosis by promoting the GSH antioxidant system and regulating iron metabolism gene expression. Nrf2 action decreases the highly reactive labile intracellular iron (LII) by upregulating the expression of iron storage (*FTL* and *FTH1*) and exporter (*FPN1*) genes [162]. The main biochemical characteristics of ferroptosis consist of increased ROS, decreased glutathione peroxidase 4 (GPX4) activity, and the accumulation of lipid metabolites [158,163]. Ferroptosis plays an important regulatory role in hematopoietic homeostasis and is closely related to the development and progression of various hematological malignancies, such as leukemia, lymphoma, and multiple myeloma [163,164,165,166,167]. Annadurai Anandhan and collaborators recently demonstrated that Nrf2 maintains iron homeostasis by controlling HERC2, the E3 ubiquitin ligase for NCOA4 and FBXL5, and the autophagosome–lysosome fusion mediator VAMP8. Nrf2 inhibition selectively enhances LII accumulation and sensitizes cancer cells and xenograft tumors to ferroptosis induction, sparing normal tissues. They state that Nrf2 does not merely have anti-ferroptotic functions, since several Nrf2 target genes, including *SLC7A11* (xCT) and *GCLM*, modulate intracellular GSH levels and thus GPX4 activity to induce ferroptosis, but the anti-ferroptosis function of Nrf2 extends to the contribution of the Nrf2-HERC2 and Nrf2-VAMP8/NCOA4 axes to control iron homeostasis and dictate ferroptosis sensitivity [168]. Wu et al., using the Nrf2 inhibitor triptolide on doxorubicin-resistant leukemic cell lines, demonstrated that Nrf2 downregulation led to leukemia cell ferroptosis [169].

The ability of Nrf2 to protect cells from ferroptosis also plays a significant role in mediating drug resistance. Several genes involved in preventing ferroptosis, including *GPX4*, *FSP1*, *HMOX1*, *SLC7A11*, *FTL*, *GCLC*, *GCLM*, *FTH1*, and the cystine/glutamate antiporter system Xc-(xCT), have been identified as Nrf2-dependent targets. Nrf2 activation can confer chemoresistance or radiation resistance by preventing the cells from undergoing ferroptosis [170,171,172,173,174]. Conversely, the inhibition of Nrf2 has been shown to increase cancer cell susceptibility to ferroptosis, potentially overcoming therapy resistance in malignancies such as acute myeloid leukemia (AML) and hepatocellular carcinoma [175]. Our recent research demonstrated that clobetasol propionate, an Nrf2 inhibitor, sensitized lung cancer cells with hyperactive Nrf2 to radiotherapy by inducing ferroptosis through a mitochondrial ROS-dependent pathway [176]. Notably, a separate study identified an alternative strategy to trigger ferroptosis in cancer cells with elevated Nrf2 activity by inhibiting the xCT transporter. This approach exploits collateral sensitivity in these cells, mediated by Nrf2-induced upregulation of the pro-ferroptotic target *ABCC1*. When xCT is inhibited by erastin, *ABCC1* expression contributes to the depletion of glutathione (GSH), thereby promoting ferroptosis [177]. Future research should focus on further elucidating the context-specific interactions between Nrf2, ferroptosis, and therapeutic resistance. This includes identifying biomarkers for predicting responsiveness and optimizing combination strategies to overcome resistance in cancers with hyperactive Nrf2.

## 7. Nrf2 Inhibitors

The enumeration above emphasizes the need to inhibit Nrf2 activity in cancer cells for optimal therapeutic outcomes. Based on the different modalities through which Nrf2 is regulated, different approaches can be employed for targeting Nrf2.

Nrf2 is known to heterodimerize with sMAF to bind to ARE. Inhibiting the domains involved in sMAF/Nrf2 interactions is one of the strategies to inhibit Nrf2. In a quantitative high-throughput screening study using around 400,000 small molecules, Singh et al. identified ML385 as an inhibitor of Nrf2 and of the expression of downstream genes [178]. ML385 binds to the Neh1 domain of Nrf2 and interferes with the binding with sMAF and hence binds to ARE sequences in DNA and transactivation. ML385 could increase the cytotoxicity of chemotherapeutic drugs, and this ability is dependent on KEAP1 mutation, implying the specificity of action of ML385. But, ML385 has also been shown to reduce the levels of Nrf2 in the cells, implying other mechanisms of action [179], and further studies are required to develop this molecule as an Nrf2-targeting drug in clinic settings. Similarly, a heterodimeric nuclear receptor consisting of retinoid-X-receptor α (RXR-α) and retinoic acid receptor-α (RAR-α) has been shown to interact with the Neh7 domain of Nrf2, thereby decreasing the expression of Nrf2-dependent genes (Figure 1) [180], sensitizing the cells to chemotherapeutic drugs. All-trans retinoic acid, a vitamin A metabolite that is an agonist of this RXR-α, induces the expression of RXR-α, acting as a potent inhibitor of Nrf2.

Apart from the direct inhibition of Nrf2, its upstream and downstream regulatory proteins have also been targeted. Brusatol—a natural compound isolated from Brucea javanica—has been identified as an inhibitor of Nrf2 by promoting the ubiquitination of Nrf2 and its degradation [181]. Consequently, the brusatol and cisplatin treatment led to a substantial increase in the apoptosis and tumor growth of A549 xenografts as compared to the cisplatin treatment alone. But further reports have indicated that the Nrf2 inhibitory effect exhibited by brusatol could be due to the impact of brusatol on the overall protein synthesis. Hence, the effect of brusatol is not specific to Nrf2, but to many short-lived proteins. In a similar line, Tsuchida et al. have identified halofuginone as a potential inhibitor of Nrf2 through screening [182]. A halofuginone treatment led to the rapid depletion in the levels of Nrf2 and ameliorated the resistance of lung cancer cells against many chemotherapeutic drugs. This depletion of Nrf2 is due to amino acid starvation and, hence, lacks the specificity that will lead to increased side effects. Another natural compound, trigonelline, an alkaloid found in coffee, has also been shown to reduce the basal and inducible (by tBHQ) nuclear levels of Nrf2, and thereby, trigonelline could break the chemoresistance of pancreatic cancer cells [183].

Since Nrf2 levels are regulated through the PI3K/AKT axis, inhibitors of this pathway can also inhibit Nrf2. Apigenin, a flavonoid compound, has been shown to reduce the levels of mRNA and protein of Nrf2 through the downregulation of the PI3K/AKT pathway, and thereby sensitizing cancer cells that are resistant to doxorubicin by inducing apoptosis [184]. Other known inhibitors of this pathway, like LY294002 and Rapamycin, can be potential inhibitors of Nrf2 activity. Similarly, the inhibitors of oncogeneic KRAS can also be potential inhibitors of Nrf2, as KRAS is known to support Nrf2 transcription [185].

Nrf2 is also subjected to GSK3β-TcRP-based proteasomal degradation in a Keap1-independent way. Enhancing this axis can have Nrf2 inhibitory effect. Lee et al. have screened 644 phytochemicals and identified Convallatoxin—a glycoside extracted from *Convallaria majalis*—as a novel inhibitor of Nrf2 [186]. Convallatoxin suppressed Nrf2 by promoting proteasomal degradation through the GSK3β-TcRP axis that requires an intact Neh6 domain of Nrf2. Further, this compound sensitized lung cancer cells to 5-fluorouracil-mediated apoptosis. In another drug screening using 4000 clinical compounds, aimed at identifying Nrf2 inhibitors, clobetasol propionate (CP) was found to be a potent inhibitor of Nrf2 [187]. CP promoted the β-TrCP-dependent degradation of Nrf2 in a glucocorticoid receptor-dependent manner, and when combined with Rapamycin treatment, CP strongly inhibited the growth of Keap1-mutated tumor cells [187] (Table 2).

Similar to upstream effectors, targeting downstream Nrf2-dependent genes can also be a suitable strategy to suppress the effect of Nrf2. Nrf2 is an important regulator of the downstream pathways, like the glutathione pathway, thioredoxin pathway, cystine uptake, NADPH synthesis through the pentose phosphate pathway, and other metabolic intermediaries. Inhibiting these crucial pathway genes can be a valuable strategy to reduce the effect of Nrf2 hyperactivation in cells. We used this strategy to inhibit thioredoxin reductase in lung cancer cells. We used curcumin derivatives, namely dimethoxy curcumin and mitocurcumin, to inhibit thioredoxin reductase 1 (cytosolic) and thioredoxin reductase 2 (mitochondrial), respectively, and achieved an effective killing of cancer cells and radiosensitization [197,198]. Beyond direct inhibition, advances in understanding Nrf2′s post-translational modifications, such as acetylation by p300/CBP, offer new avenues [27]. While acetylation typically enhances Nrf2 activity, disrupting this process by p300/CBP inhibitors could suppress Nrf2 function in cancer cells, thereby achieving therapy sensitization.

Despite significant progress in the development of Nrf2 inhibitors, no molecule has yet received clinical approval as a specific Nrf2 inhibitor, highlighting the critical need for continued research in this field. The currently identified Nrf2 inhibitors are mostly based on the random screening of chemical libraries using in vitro screening methods. These molecules often lack sufficient specificity and exhibit considerable off-target effects, which complicates their therapeutic application. It is important to develop inhibitors based on certain rational strategies using the structural information of Nrf2 and KEAP1. This approach may result in more specific inhibitors of Nrf2. To address these challenges, novel strategies are under investigation to develop more precise and effective Nrf2 inhibitors. These include small peptide molecules and stapled peptides (short α-helical peptides stabilized by a hydrocarbon staple) designed to directly disrupt Nrf2 interactions, such as its binding to DNA or its interaction with small Maf (sMaf) proteins [199,200]. These approaches aim to enhance specificity by targeting critical protein–protein or protein–DNA interfaces unique to Nrf2 signaling. Furthermore, advanced protein-targeting mechanisms, such as Proteolysis-Targeting Chimeras (PROTACs), are being explored to selectively degrade Nrf2, offering a promising avenue for the precise modulation of Nrf2 activity [201].

Additionally, achieving tumor cell-specific inhibition of Nrf2 is a priority, given the essential role of Nrf2 in maintaining normal cellular functions, such as antioxidant defense and redox homeostasis [202]. To achieve the selective inhibition of Nrf2 in tumor cells while preserving its function in normal cells, innovative approaches are being developed. For instance, Aboulkassim et al. have introduced a chemical compound, R16, which functions as ‘molecular glue’. R16 specifically binds to mutated KEAP1, facilitating the interaction between mutated KEAP1 and Nrf2 [203]. This interaction promotes the targeted degradation of Nrf2 in tumor cells harboring KEAP1 mutations, thereby sparing Nrf2 in normal cells with wild-type KEAP1. The other alternate approaches could be the tagging of Nrf2 inhibitors with tumor-specific ligands so that off-target effects can be minimized.

To effectively utilize various classes of Nrf2 inhibitors in cancer therapy, the development of biomarkers is also essential for stratifying patients with Nrf2 overexpression and identifying those likely to benefit from specific inhibitors. As discussed in the article, Nrf2 hyperactivation arises from diverse molecular mechanisms, including somatic mutations in KEAP1, the oncogenic activation of KRAS, PI3K/AKT pathway dysregulation, microRNA-mediated mechanisms, and other epigenetic alterations [3]. This knowledge and understanding can be leveraged for the development and validation of multiomics-based biomarkers such as KEAP1 mutations, elevated Nrf2 mRNA/protein expression, and Nrf2-driven gene signatures. These biomarkers can facilitate patient stratification for clinical trials and further enable the development of tailored therapeutic strategies for cancer treatment.

## 8. Nrf2 in Cancer: Opportunities and Problems

Nrf2 plays a dual role in cancer, which can protect normal cells from oxidative stress but also promote tumor survival and therapy resistance in cancers like leukemia and lung and hepatocellular carcinoma. Its strength lies in its ability to maintain redox homeostasis, detoxify carcinogens, and prevent DNA damage, thereby protecting against cancer initiation. However, in advanced-stage cancers, persistent Nrf2 activation facilitates tumor progression by promoting cell survival.

### 8.1. Strengths of Nrf2 as a Target in Cancer

The strengths of targeting Nrf2 in cancer are numerous; in the following sections, we will enumerate and discuss the key advantages that make Nrf2 an attractive therapeutic target:Protection against oxidative stress: Nrf2 protects cancer cells from the high levels of oxidative stress that result from rapid cell proliferation and altered metabolism. By upregulating antioxidant genes (e.g., HO-1, NQO1, and GCLC), Nrf2 helps maintain cellular redox homeostasis, promoting cancer cell survival in tumors with high oxidative stress. In early-stage cancers, Nrf2 can contribute to DNA repair and resistance to apoptosis, thus supporting tumor growth and favoring the emergence of clones resistant to therapies [202].Chemoresistance: Nrf2 activation leads to the expression of drug-metabolizing enzymes (e.g., GST and CYP450), which detoxify chemotherapeutic agents, thus reducing the effectiveness of chemotherapy [204]. In cancers such as non-small-cell lung cancer (NSCLC) and ovarian cancer, elevated Nrf2 expression is linked to poor responses to chemotherapy drugs like cisplatin and paclitaxel [7].Support for metastasis: Nrf2 plays a role in the epithelial–mesenchymal transition (EMT), a process that allows cancer cells to acquire migratory and invasive properties. In cancers like breast, pancreatic, and prostate cancer, Nrf2 promotes metastasis by enhancing the invasive potential of tumor cells [205].Tumor survival and growth: In hypoxic tumor environments, such as those found in gliomas and pancreatic cancer, Nrf2 promotes mitochondrial function and energy production, thereby supporting tumor growth under conditions of oxidative stress [206,207].Nrf2 in late-stage cancers: The overexpression of Nrf2 in late-stage cancers (e.g., lung, colon, and liver cancers) is associated with aggressive tumor behavior, as Nrf2 helps cancer cells resist oxidative stress and chemotherapeutic agents. In these cases, Nrf2 inhibitors could reduce tumor progression and enhance the effectiveness of chemotherapy [207,208].Cancer stem cells (CSCs): Nrf2 is often involved in maintaining the stem-like properties of cancer stem cells, which are resistant to chemotherapy and contribute to tumor relapse. In cancers like breast, lung, and pancreatic cancers, Nrf2 activation may promote CSC survival and self-renewal, contributing to chemoresistance [205].Compromised antioxidant defense in certain tumors: In certain cancers, such as APL, where the Nrf2 half-life is reduced [121] or in glioblastoma and colorectal cancer, where mutations in the Nrf2 pathway impair its antioxidant defense function, these tumors become more vulnerable to oxidative damage [209]. Consequently, they may be more responsive to therapies that induce the generation of ROS, such as radiation or chemotherapy.

### 8.2. Problems of Nrf2 as a Target in Cancer

Despite promising preclinical results, challenges remain, such as the lack of clinically approved Nrf2 inhibitors, off-target effects, and the need for tumor-specific targeting to spare normal cells. Overall, while Nrf2 presents a valuable therapeutic target, its complexity necessitates further research to develop precise and effective interventions.

## 9. Conclusions

Nrf2 lies at the center of a complex regulatory network essential for cellular homeostasis, particularly in stem cells. While best known as the master regulator of the antioxidant response, Nrf2 also plays critical roles in the regulation of metabolism, mitochondrial function, and ferroptosis. In many cancers, Nrf2 activity is aberrantly elevated, contributing to drug resistance and tumor progression. Thus, targeting Nrf2 represents a promising strategy to limit cancer cell proliferation and clonal selection.

However, the development of Nrf2 inhibitors remains in its early stages. There are no Nrf2-targeting agents that have been approved for clinical use, largely due to the intrinsic challenges of inhibiting a protein that lacks a well-defined active site or deep binding pocket. The lack of specific Nrf2 inhibitors is also due to the non-availability of the Nrf2 crystal structure. In addition, existing compounds suffer from poor specificity and toxicity to normal tissues. Therefore, continued research efforts are needed to overcome these limitations and develop effective, selective therapeutics against Nrf2-overexpressing tumors.

## Figures and Tables

**Figure 1 cells-14-00713-f001:**
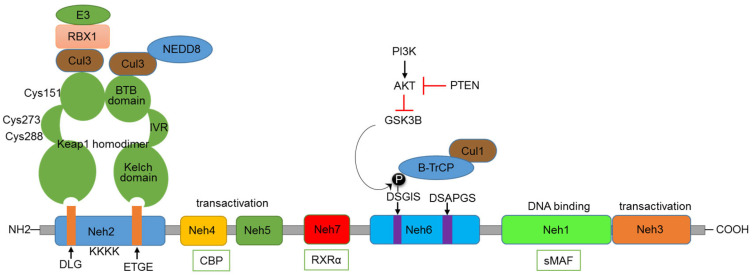
Domain structure of Nrf2 and major regulatory mechanisms.

**Figure 2 cells-14-00713-f002:**
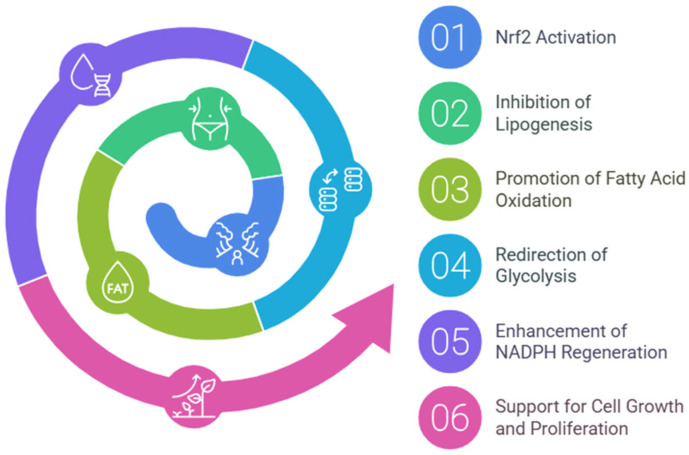
Nrf2-mediated metabolic reprogramming. This diagram illustrates the key metabolic pathways regulated by Nrf2, highlighting its role in energy metabolism.

**Figure 3 cells-14-00713-f003:**
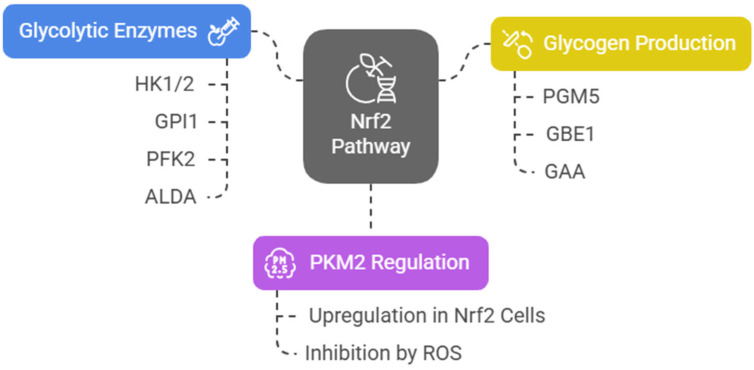
NRF2 pathway and glycolytic regulation.

**Figure 4 cells-14-00713-f004:**
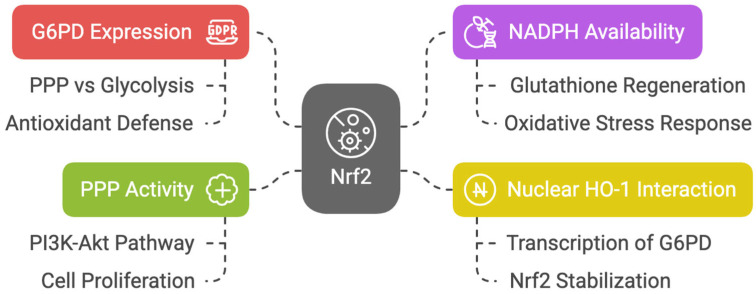
Diagram about Nrf2′s role in PPP and antioxidant defense.

**Figure 5 cells-14-00713-f005:**
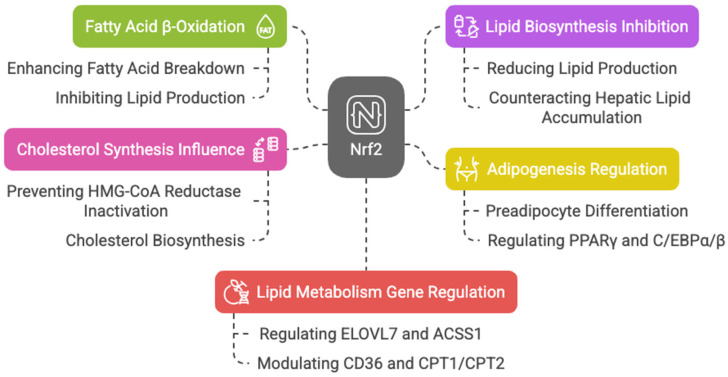
Diagram about Nrf2′s role in lipid metabolism and energy production.

**Figure 6 cells-14-00713-f006:**
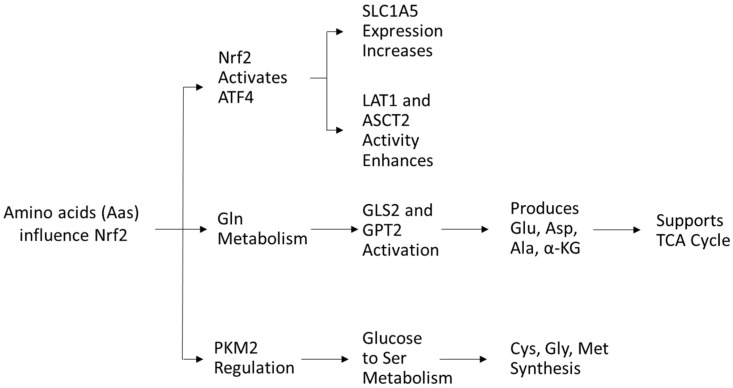
Diagram about amino acids and Nrf2 regulation.

**Figure 7 cells-14-00713-f007:**
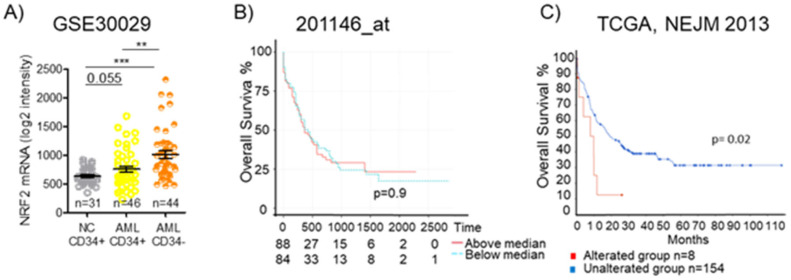
Nrf2 expression and overall survival in AML. (**A**) Expression of NFE2L2 (Nrf2) in CD34+ and CD34- fractions from BM of healthy donors (NC) or AML patients as detected by microarray in GSE30029 dataset. Statistical analysis was performed using the Student’s *t*-test, ** *p* ≤ 0.005, *** *p* ≤ 0.0005. (**B**) Kaplan–Meier survival analyses of AML patients based on their Nrf2 expression level. Overall survival in TCGA AML dataset for 201146at. (**C**) Kaplan–Meier survival analyses of AML patients based on the presence or absence of Nrf2 alterations in The Cancer Genome Atlas (TCGA), NEJM 2013. OSm, medium overall survival. Survival analysis based on log-rank test (*p* < 0.05).

**Figure 8 cells-14-00713-f008:**
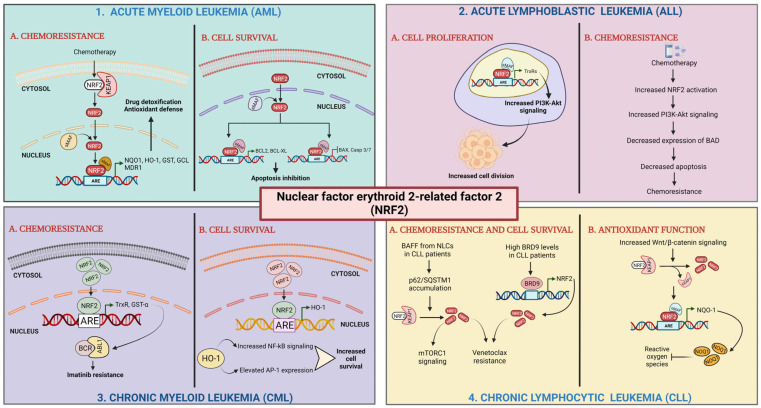
Role of Nrf2 in leukemia. (**1**) Acute myeloid leukemia: (**1A**) chemoresistance; Nrf2 activation by chemotherapy results in its translocation to nucleus, heterodimer formation with sMAF proteins, and transactivation of AREs leading to high expression of detoxifying and antioxidant enzymes and drug efflux transporter. (**1B**) Cell survival; elevated Nrf2 levels activate the transcription of antiapoptotic genes and repress the transcription of pro-apoptotic genes [100,124]. (**2**) Acute lymphoblastic leukemia (ALL): (**2A**) cell proliferation; Nrf2 activates the expression of antioxidant enzymes leading to enhanced PI3K-Akt signaling. (**2B**) Chemoresistance; Nrf2 activation upregulates PI3K-Akt signaling. Enhanced PI3K-Akt activation represses BAD expression [129,132]. (**3**) Chronic myeloid leukemia (CML): (**3A**) chemoresistance; elevated Nrf2 levels upregulate the expression of antioxidant enzymes. (**3B**) Cell survival; high levels of Nrf2 causes upregulation of HO-1 expression [127,128]. (**4**) Chronic lymphocytic leukemia (CLL): (**4A**) chemoresistance and cell survival; BAFF released from NLCs present in the tumor microenvironment enhances the accumulation of p62/SQSTM1, which triggers Nrf2 activation. High BRD9 levels transcriptionally activate Nrf2 expression. (**4B**) Antioxidant function; increased Wnt/β-catenin levels cause Nrf2 activation stimulating expression of NQO-1 by binding to ARE sequences [135].

**Table 1 cells-14-00713-t001:** Nrf2 effects in leukemia.

Type of Leukemia	Nrf2 Stimulation (↑)/Inhibition (↓) Activity	References
Acute myeloid leukemia (AML)	Arsenic ↑ Nrf2 translocation to nucleus; ↓ ROS production; ↑ expression of antioxidant enzymes	[100]
	↑ Nrf2 expression; ↑ resistance to chemotherapy	[122]
	Vitamin D activates Nrf2; ↑ myeloid differentiation	[123]
	NF-κB ↑ Nrf2; ↑ increases oncogenic cell proliferation and survival; ↑ chemoresistance	[124]
Acute promyelocytic leukemia (APL)	↓ Nrf2 and its target genes led to ↑ sensitivity to oxidative stress therapy, such as ascorbic acid	[121,138]
Acute Lymphocytic Leukemia (ALL)	Mutations in Nrf2/Keap1 pathways (73% of pediatric ALL patients)	[129]
	Elevated Nrf2 expression ↓ PAQR3;↑ ALL progression	[130]
	Nrf2 signaling inhibition by brusatol ↑ ROS and O_2_^−^ and apoptosis of ALL cells	[131]
	Nrf2 overexpression ↑ PI3-AKT signaling; ↓ BAD response to chemotherapy	[132]
Chronic myeloid leukemia (CML)	Nrf2 targets HO-1 and NQO1↓ apoptosis; ↑ resistance to imatinib	[126]
	↓ Nrf2; ↑ ROS and imatinib induced apoptosis	[127]
	Inhibitors of Nrf2 ↓ GST-α; ↑ 4HNE and sensitivity toward imatinib	[128]
Chronic lymphocytic leukemia (CLL)	↑ Nrf2 signaling in CLL; electrophilic and antioxidant compounds;↓ Nrf2 signaling; ↑ CLL-selective cytotoxicity	[135]
	Cross talk between NF-κB signaling and Nrf2 via p62/mTORC1; ↑ NQO1 and ROR1; ↓ response to ROS-inducing therapy	[136]

**Table 2 cells-14-00713-t002:** List of Nrf2 inhibitors and their mechanism of action.

Nrf2 Inhibitors	Mechanism of Action	Reference
ML385	Inhibition of transactivation of Nrf2	[178]
malabaricone-A	Inhibition of transactivation of Nrf2	[188]
AEM1	Inhibition of transactivation of Nrf2	[189]
brusatol	Inhibition of protein synthesis	[181]
clobetasol propionate	Ligand of glucocorticoid receptor	[187]
dexamethasone	Ligand of glucocorticoid receptor	[190]
bexarotene	Agonists of the retinoic acid receptor-α	[180]
all-trans retinoic acid	Agonists of the retinoic acid receptor-α	[191]
wogonin	Impacting the stability of Nrf2 transcript	[192]
luteolin	Impacting the stability of Nrf2 transcript	[193]
ochratoxin A	Prevents nuclear translocation	[194]
trigonelline	Prevents nuclear translocation	[183]
halofuginone	Decreased Nrf2 protein	[182]
pyrazolyl hydroxamic acid	Pyrazolyl hydroxamic acid	[195]
ascorbic acid	Reduced Nrf2/ARE complex	[196]

## Data Availability

Not applicable.

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
