# Peer review of "Molecular Targets of Oxidative Stress: Focus on Nuclear Factor Erythroid 2–Related Factor 2 Function in Leukemia and Other Cancers"

_cells, 2025, doi:10.3390/cells14100713_

Round 1
Reviewer 1 Report
Comments and Suggestions for Authors
The manuscript entitled: “Molecular Targets of Oxidative Stress: Focus on the Nrf2 function in Leukemia and Cancer (ID: cells-3615786)” by Syed et al. aims to provide an overview of Nrf2 regulation and function, especially in cancer biology and its role in metabolic adaption and drug resistance, Nrf2`s role in leukemia and potential strategies for its therapeutic options.
Albeit the review is well written and of special interest, comments should be addressed to further improve the manuscript.
Comments:
- Section 7: please provide more detailed information how these potential therapeutic modulations could be addressed in future studies. Moreover, please highlight more deeply the unmet clinical need of NRf2 inhibitors and what kind of patient would profit from this therapeutic option.
- A kind of discussion is missing about the strength and weaknesses of Nrf2 in different cancers and its potential therapeutic options.
- Figure 2, 3 and 7 should be enlarged. Figure 6: please provide abbreviations where appropriate. Figure 8: Please provide a reference for this within the figure legend.
Author Response
Reviewer 1
Q1: Section 7: please provide more detailed information how these potential therapeutic modulations could be addressed in future studies. Moreover, please highlight more deeply the unmet clinical need of NRf2 inhibitors and what kind of patient would profit from this therapeutic option.
A1: We thank the reviewers for this valuable comment. More detailed information has been provided in the revised Section 7 (page 23), including potential therapeutic strategies, unmet clinical needs, and patient populations that may benefit from Nrf2-targeted therapies.
Q2: A kind of discussion is missing about the strength and weaknesses of Nrf2 in different cancers and its potential therapeutic options
A2: We thank the reviewers for their valuable comment. To address this point, we have added a new Section 8 (page 24-25) discussing the strengths and weaknesses of Nrf2 in various cancers and its potential therapeutic relevance.
Q3: Figure 2, 3 and 7 should be enlarged. Figure 6: please provide abbreviations where appropriate. Figure 8: Please provide a reference for this within the figure legend.
A3: We thank the reviewers for these helpful suggestions. Figures 2, 3, and 7 have been enlarged to improve clarity. Abbreviations have been added to Figure 6 where appropriate, and a reference has been included in the legend of Figure 8.

Reviewer 2 Report
Comments and Suggestions for Authors
This comprehensive and well-organized review addresses the dual role of Nrf2 in redox regulation and cancer progression, with a strong and timely focus on leukemia. The manuscript effectively summarizes current knowledge on Nrf2-driven metabolic reprogramming, chemoresistance, ferroptosis, and leukemia-specific mechanisms, supported by extensive referencing.
The review is scientifically sound and of interest to Cells readers. I recommend minor revisions to improve clarity and flow:
- Redundancy: Some mechanistic content (e.g., HO-1, NQO1) is repeated across sections. Consolidation is suggested.
- Transitions: Enhance the flow between major sections, particularly between metabolism and mitochondrial regulation.
- Epigenetics/miRNA: Expand slightly on the role of miRNAs and KEAP1 promoter methylation in leukemia.
- Clinical Relevance: A brief mention of ongoing Nrf2-targeted strategies and potential biomarkers would strengthen translational value.
- Language: Several typographical and formatting issues (e.g., hyphenation, line breaks, grammar) should be corrected.
Small errors should be corrected.
Author Response
Reviewer 2
Q1: Redundancy: Some mechanistic content (e.g., HO-1, NQO1) is repeated across sections. Consolidation is suggested.
A1: We have carefully reviewed all instances where HO-1 and NQO1 were discussed and have amended the manuscript to remove redundancy. The revised version consolidates the mechanistic content to improve clarity and flow.
Q2: Transitions: Enhance the flow between major sections, particularly between metabolism and mitochondrial regulation
A2: We thank the reviewers for this constructive suggestion. To improve the flow between major sections, particularly between metabolism and mitochondrial regulation, a transitional paragraph has been added at the beginning of Section 4 (page 9).
Q3: Epigenetics/miRNA: Expand slightly on the role of miRNAs and KEAP1 promoter methylation in leukemia.
A3: Thank you for the suggestion. In response, we have added a concise paragraph on page 4 discussing the role of miRNAs and KEAP1 promoter methylation in leukemia, highlighting how these mechanisms contribute to dysregulation of the KEAP1-NRF2 axis and may promote chemoresistance.
Q4: Language: Several typographical and formatting issues (e.g., hyphenation, line breaks, grammar) should be corrected. Small errors should be corrected.
A4: Thank you for bringing this to our attention. We have now thoroughly reviewed the manuscript and corrected all typographical, formatting, and grammatical errors to improve overall clarity and presentation.
